# An $O(\log_2 N)$ Fully-Balanced Resampling Algorithm for Particle Filters on Distributed Memory Architectures

**Alessandro Varsi** [1,*], **Simon Maskell** [1] **and Paul G. Spirakis** [2,3]

1   Department of Electrical Engineering and Electronics, University of Liverpool, Liverpool L69 3GJ, UK; S.Maskell@liverpool.ac.uk
2   Department of Computer Science, University of Liverpool, Liverpool L69 3BX, UK; spirakis@liverpool.ac.uk
3   Department of Computer Engineering and Informatics, University of Patras, 26504 Patras, Greece
*   Correspondence: Alessandro.Varsi@liverpool.ac.uk

**Abstract:** Resampling is a well-known statistical algorithm that is commonly applied in the context of Particle Filters (PFs) in order to perform state estimation for non-linear non-Gaussian dynamic models. As the models become more complex and accurate, the run-time of PF applications becomes increasingly slow. Parallel computing can help to address this. However, resampling (and, hence, PFs as well) necessarily involves a bottleneck, the redistribution step, which is notoriously challenging to parallelize if using textbook parallel computing techniques. A state-of-the-art redistribution takes $O((\log_2 N)^2)$ computations on Distributed Memory (DM) architectures, which most supercomputers adopt, whereas redistribution can be performed in $O(\log_2 N)$ on Shared Memory (SM) architectures, such as GPU or mainstream CPUs. In this paper, we propose a novel parallel redistribution for DM that achieves an $O(\log_2 N)$ time complexity. We also present empirical results that indicate that our novel approach outperforms the $O((\log_2 N)^2)$ approach.

**Keywords:** parallel computing; resampling; Particle Filters; high performance computing; Distributed Memory; message passing interface

## 1. Introduction

### 1.1. Motivation

In several modern applications, it is often necessary to estimate the state of a system, given a mathematical model for the system and a stream of noisy observations. Particle Filters (PFs) are typically used in this context. The key idea is to sample $N$ hypotheses (i.e., particles) from an arbitrary proposal distribution to approximate the probability density function (pdf) of the true state. However, at some point, the particles experience a numerical error, called particle degeneracy, which makes the estimates diverge from the true state. A *resampling* algorithm is then applied to correct for degeneracy by replacing the particles that are diverging from the true state with copies of the particles that are not doing so [1]. This sampling–resampling approach is highly flexible, such that PFs find application in a wide range of fields, ranging from machine learning [2] to medical research [3], fault prediction [4], weather forecasting [5], tracking [6] or, broadly speaking, any domain involving decision making in response to streaming data. Resampling is also used in other Monte Carlo methods, such as sequential Monte Carlo samplers [7,8] and PHD filters [9]. For brevity, this paper focuses exclusively on PF contexts.

Modern efforts of making models more detailed have translated to an increasing demand in making PFs more accurate. This demand can be satisfied in several ways, ranging from applying better proposal distributions [10] to collecting more measurements [11] and using more particles [12,13]. Using more particles is especially important in settings where we are more interested in computing the probability that the true state falls within a certain state-space region, rather than simply estimating its mean [14]. However, the likely side-effect of any of these approaches is a significant increment to the run-time, and this

may be critical, especially in real-time applications [12]. Parallel computing becomes necessary in order to compensate for this side-effect.

*1.2. Problem Definition and Related Work*

Although the particles can be sampled in embarrassingly parallel fashion, resampling is hard to parallelize *globally*, i.e., in such a way that the result for $P > 1$ processing units (or cores, as referred to in this paper) is identical to that achieved using a single core. This is because of the difficulties in parallelizing the constituent *redistribution* step, whose textbook implementation achieves an $O(N)$ time complexity on one core. In order to bypass the need to parallelize resampling, one could use multiple PFs in parallel, each performing resampling locally, i.e., when each core performs resampling independently without considering the content of the other cores' particles. However, this approach has shown accuracy, scalability and model-dependent applicability issues [15–17].

On Shared Memory (SM) architectures, it has been shown that a global redistribution of the particles takes $O(\log_2 N)$ computations by using a static load balancing approach, in which, all cores perform independently up to $N$ binary searches in order to achieve a perfect workload balance. Examples are found in [12,13,18,19] on GPU and mainstream CPUs. High Performance Computing (HPC) applications, however, need to use Distributed Memory (DM) architectures to overcome the limitations in modern SM of a low memory capacity and Degree Of Parallelism (DOP).

On DM, parallelization is more complicated, as the cores cannot directly access the other cores' memory without exchanging messages. Three master–worker solutions for DM (along with mixed versions of them) are presented in [20]: Centralized Resampling (C-R), Resampling with Proportional Allocation (RPA) and Resampling with Non-proportional Allocation (RNA). C-R performs resampling globally, but here, a central-unit gathers the particles from all cores, performs redistribution sequentially and scatters the result back to the cores, making this algorithm scale as $O(N)$. RPA also performs global resampling, but the network topology is randomized, as the redistribution is partly or potentially entirely (worst-case) centralized to one or a few cores, leading to a strongly data-dependent run-time and an $O(N)$ time complexity in the worst-case. RNA has a simpler central unit and communication pattern than RPA, but sacrifices accuracy, as local resampling is performed. For the routing, RNA could use RPA or a ring topology, where the cores cyclically exchange a user-defined number of particles with their neighbors, though such an approach risks redundant communication. These master–worker approaches have been used, re-interpreted or mixed in recent work, such as in [17,21–24]. In [17,24,25], it is shown that such strategies may have accuracy or scalability issues, especially for a highly unbalanced workload, large $N$ or DOP. In this paper, we then consider only *fully-balanced* solutions, which we define as follows.

**Definition 1.** *A fully-balanced redistribution meets the following requests:*
- *All cores perform the same pre-agreed tasks (i.e., no central unit(s) are involved) to balance the workload evenly;*
- *The number of messages for the load balancing is data-independent in order to guarantee a stable run-time, as often required in real-time applications;*
- *The redistribution of the particles is performed globally in order to ensure the same output of sequential redistribution and that no speed–accuracy trade-off is made when the DOP increases.*

In [26], it has been shown that redistribution can be parallelized in a fully-balanced fashion on DM by using a divide-and-conquer approach that recursively sorts and splits the particles. Since Bitonic Sort [27–29] is performed $O(\log_2 N)$ times, the achieved time complexity is $O((\log_2 N)^3)$. In [17], sort is also employed recursively in a dynamic scheduler for RPA/RNA. In [30], the time complexity is reduced to $O((\log_2 N)^2)$ by showing that Bitonic Sort is only needed once. In [25], this new idea is ported from MapReduce to the Message

Passing Interface (MPI) and then further optimized in [7]. However, the data movement in this fully-balanced redistribution is still the bottleneck, especially for high DOP.

*1.3. Our Results*

This paper proposes a novel fully-balanced approach for DM that achieves an $O(\log_2 N)$ time complexity and improves on the redistribution algorithms described in [7,25]. Our experimental results demonstrate that our novel redistribution is approximately eight times faster than the $O((\log_2 N)^2)$ ones on a cluster of 256 cores.

The rest of the paper is organized as follows: in Section 2, we briefly describe PFs, with a particular emphasis on resampling (and redistribution). In Section 3, we give brief information about DM and MPI. In Section 4, we describe our novel parallel redistribution in detail and include proof of its algorithmic complexity. In Section 5, we show the numerical results for redistribution first, and then for a PF example. In Section 6, we outline our conclusions and give recommendations for future work.

## 2. Sequential Importance Resampling

In this section, we briefly describe Sequential Importance Resampling (SIR), one of the most common variants of PFs [1].

PFs employ the Importance Sampling (IS) principle to estimate the true state $\mathbf{X}_t \in \mathbb{R}^M$ of a system. IS uses an arbitrary proposal distribution $q(\mathbf{X}_t|\mathbf{X}_{t-1})$ to randomly generate $\mathbf{x}_t \in \mathbb{R}^{N \times M}$, a population of $N$ statistically independent hypotheses for the state, called particles. Each particle $\mathbf{x}_t^i$ is then weighted by an unnormalized importance weight $\mathbf{w}_t^i \in \mathbb{R}$ $\forall i = 0, 1, ..., N-1$, which is computed based on the resemblance between $\mathbf{x}_t^i$ and $\mathbf{X}_t$. In this way, $\mathbf{x}_t$ approximates the posterior of the true state.

To weight each particle correctly, at every time step $t$, we collect $\mathbf{Y}_t \in \mathbb{R}^{M_y}$, a set of measurable quantities that are related to $\mathbf{X}_t$ by some pdf. At the initial time $t = 0$, no measurement has been collected, so the particles are initially drawn from the initial distribution $q_0(\mathbf{X}_0) = p_0(\mathbf{X}_0)$ and weighted equally as $1/N$. Then, for any time step $t = 1, 2, ..., T_{PF}$, measurements are collected, and each particle is drawn from the proposal distribution as follows:

$$\mathbf{x}_t^i \sim q(\mathbf{x}_t^i|\mathbf{x}_{t-1}^i, \mathbf{Y}_t) \tag{1}$$

and weighted by

$$\mathbf{w}_t^i = \mathbf{w}_{t-1}^i \frac{p(\mathbf{x}_t^i|\mathbf{x}_{t-1}^i)p(\mathbf{Y}_t|\mathbf{x}_t^i)}{q(\mathbf{x}_t^i|\mathbf{x}_{t-1}^i, \mathbf{Y}_t)} \tag{2}$$

where $p(\mathbf{x}_t^i|\mathbf{x}_{t-1}^i)$ and $p(\mathbf{Y}_t|\mathbf{x}_t^i)$ are known from the model. The IS step performs (1) and (2) in sequence. Then, to effectively represent the pdf of the state, the weights are normalized as follows:

$$\tilde{\mathbf{w}}_t^i = \mathbf{w}_t^i / \sum_{z=0}^{N-1} \mathbf{w}_t^z \tag{3}$$

However, the side effect of using IS is degeneracy, a phenomenon that (within a few iterations) makes all weights but one decrease towards 0. The variance of the weights is indeed proven to increase at every $t$ [1]. The most popular strategy for tackling degeneracy is to perform resampling, a task that regenerates the particle population by deleting the particles with low weights and duplicating those with high weights. In SIR, resampling is performed if the Effective Sample Size (ESS)

$$N_{eff} = 1 / \sum_{i=0}^{N-1} (\tilde{\mathbf{w}}_t^i)^2 \tag{4}$$

decreases below an arbitrary threshold, which is commonly set to $\frac{N}{2}$.

Several resampling schemes exist [31] and can be described as comprising three steps. The first step is to process the normalized weights $\tilde{\mathbf{w}}$ to generate $\mathbf{ncopies} \in \mathbb{Z}^N$ whose $i$-th element, $\mathbf{ncopies}^i$, defines how many times the $i$-th particle must be copied, such that

$$\sum_{i=0}^{N-1} \mathbf{ncopies}^i = N \tag{5}$$

The second step is redistribution, which involves duplicating each particle $\mathbf{x}^i$ as many times as $\mathbf{ncopies}^i$. A textbook redistribution is described by Algorithm 1, which we refer to as Sequential Redistribution (S-R). This algorithm achieves an $O(N)$ time complexity on a single core as (5) holds. A practical example for $N = 8$ particles is shown in Figure 1. In the final step of resampling, all weights are reset to $\frac{1}{N}$.

---

**Algorithm 1** Sequential Redistribution (S-R)

---

**Input:** $\mathbf{x}$, $\mathbf{ncopies}$, $N$
**Output:** $\mathbf{x}_{new}$

1: $z \leftarrow 0$
2: **for** $j \leftarrow 0; j < N; j \leftarrow j + 1$ **do**
3: 　　**for** $k \leftarrow 0; k < \mathbf{ncopies}^j; k \leftarrow k + 1$ **do**
4: 　　　　$\mathbf{x}_{new}^z \leftarrow \mathbf{x}^j, z \leftarrow z + 1$
5: 　　**end for**
6: **end for**

---

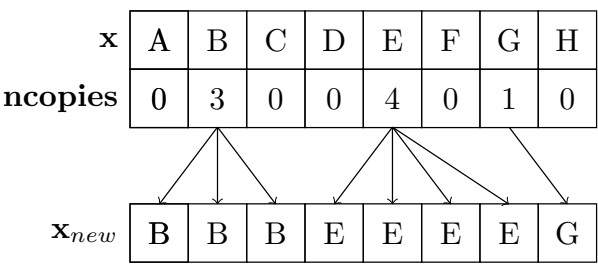

**Figure 1.** S-R—example for $N = 8$. Each $\mathbf{x}^i$ is actually a real vector, but is marked with a capital letter for brevity.

We note that all resampling schemes require redistribution, independently of the chosen strategy to perform the first step. Since we focus on proposing a novel fully-balanced redistribution, to perform the first step, we focus on using systematic resampling [31], which is also known as Minimum Variance Resampling (MVR) [7,25,26,30]. The key idea of MVR is to first compute the cumulative density function of $\tilde{\mathbf{w}}_t$, $\mathbf{cdf} \in \mathbb{R}^{N+1}$, as follows:

$$\mathbf{cdf}^i = \sum_{z=0}^{i-1} \tilde{\mathbf{w}}^z \quad \forall i = 0, 1, 2, \ldots, N \tag{6}$$

a random variable $u \sim \mathtt{Uniform}[0, 1)$ is then sampled, such that each $\mathbf{ncopies}^i$ can then be calculated as follows:

$$\mathbf{ncopies}^i = \lceil \mathbf{cdf}^{i+1} - u \rceil - \lceil \mathbf{cdf}^i - u \rceil \tag{7}$$

where the brackets represent the ceiling function (e.g., $\lceil 3.3 \rceil = 4$).

After resampling, a new estimate of $\mathbf{X}_t$ is computed as

$$\Xi_t = E(\mathbf{X}_t) = \sum_{i=0}^{N-1} \mathbf{w}_t^i \mathbf{x}_t^i \tag{8}$$

## 3. Distributed Memory Architectures

DM is a type of parallel system that is inherently different to SM. In this environment, the memory is distributed over the cores and each core can only directly access its own private memory. The exchange of information stored in the memory of the other cores is achieved

by sending/receiving explicit messages through a common communication network (see Figure 2).

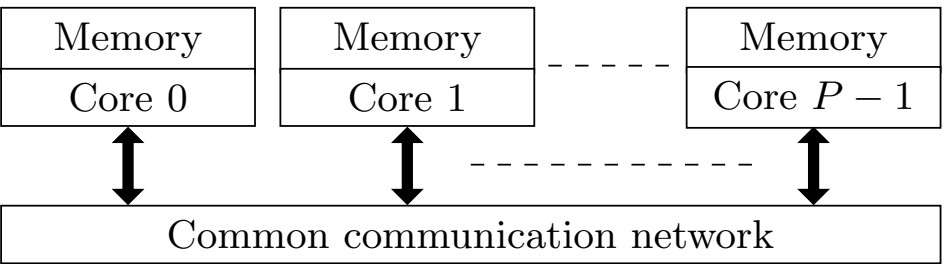

**Figure 2.** Distributed memory architecture.

DM provides several advantages over SM, such as: scalable and larger DOP; scalable and larger memory; memory contention (and other issues that stem from multiple cores accessing the same addresses in memory) not being a concern on DM. The main disadvantage of DM is the cost of communication and the consequent data movement. This can affect the speed-up relative to a single core implementation.

Any Application Programming Interface (API) for DM could be used. We choose MPI, since its intuitive syntax means it is arguably the most popular API for DM. MPI is also used in several referenced works on parallel redistribution for DM [7,17,22,23,25]. In MPI, the $P$ cores are uniquely identified by a rank $p = 0, 1, ..., P - 1$, connected via communicators, and use explicit communication routines, e.g., `MPI_Sendrecv`, to exchange messages of arbitrary size.

Most algorithms that we implement in this paper use the divide-and-conquer paradigm. Therefore, we recommend using a power-of-two number of cores to balance the communication between them. For the same reason, the number of particles $N$ should also be a power-of-two. In this case, the particles **x** and every array related to them, e.g., **w**, **ncopies** etc., can then be evenly split between the cores. Every core then always owns exactly $n = \frac{N}{P}$ elements of all arrays, whose global indexes are spread over the cores in increasing order. This means that, given a certain $N$, $P$ pair, the $i$-th particle (where $i \in [0, N - 1]$) will always belong to the same core with rank $p = \left\lfloor \frac{i}{n} \right\rfloor$. More precisely, while all equations in this paper use global indexes for simplicity, each core $p$ actually uses a local index $j \in [0, n - 1]$ to address arrays and achieve parallelism, knowing that each $j$ corresponds to global index $i = j + pn$. The space complexity is then $O(\frac{N}{P})$ for any $P \leq N$.

## 4. Novel $O(\log_2 N)$ fully-balanced Redistribution

In this section, we prove how it is possible to redistribute $N$ particles in $O(\log_2 N)$ parallel time on DM by using a novel fully-balanced redistribution algorithm, which we name Rotational Nearly Sort and Split (RoSS) redistribution. We also provide details on how to implement RoSS on MPI. We refer any reader who is interested in the description of the resulting algorithms to Algorithms 2–5.

---

**Algorithm 2** Rotational Nearly Sort

---

**Input:** $\mathbf{x}$, **ncopies**, $N$, $P$, $n = \frac{N}{P}$, $p$
**Output:** $\mathbf{x}$, **ncopies**

1: $\mathbf{x}$, **ncopies**, **zeros** $\leftarrow$ S-NS$(\mathbf{x}, \mathbf{ncopies}, n)$, see Algorithm 3
2: **shifts** $\leftarrow$ `Exclusive_Cumulative_Sum`(**zeros**)
3: **if** $P < N$ **then** perform leaf stage of the binary tree
4:      *partner* $\leftarrow (p-1)$ & $(P-1)$, i.e., the neighbor
5:      **if shifts** & $(n-1) > 0$ **then**
6:          **for** $j \leftarrow 0; j < n; j \leftarrow j+1$ **do**
7:              **if** $j <$ **shifts** & $(n-1)$ **then**
8:                  Send $\mathbf{x}^j$, **ncopies**$^j$ to *partner*, **ncopies**$^j \leftarrow 0$
9:              **else**
10:                  Shift particle to the left by **shifts** & $(n-1)$
11:              **end if**
12:          **end for**
13:          **shifts** $\leftarrow$ **shifts** $-$ **shifts** & $(n-1)$
14:          Send **shifts** to *partner*
15:      **else**
16:          Send arrays of 0s to *partner* (Message to reject)
17:      **end if**
18:      Accept or reject the received particles and **shifts**
19: **end if**
20: **for** $k \leftarrow 1; k \leq \log_2 P; k \leftarrow k+1$ **do** binary tree
21:      *partner* $\leftarrow (p - 2^{k-1})$ & $(P-1)$
22:      **if shifts** & $n2^{k-1} > 0$ **then**
23:          **for** $j \leftarrow 0; j < n; j \leftarrow j+1$ **do**
24:              Send $\mathbf{x}^j$, **ncopies**$^j$ to *partner*, **ncopies**$^j \leftarrow 0$
25:          **end for**
26:          **shifts** $\leftarrow$ **shifts** $-$ **shifts** & $n2^{k-1}$
27:          Send **shifts** to *partner*
28:      **else**
29:          Send arrays of 0s to *partner* (Message to reject)
30:      **end if**
31:      Accept or reject the received particles and **shifts**
32: **end for**

---

**Algorithm 3** Sequential Nearly Sort (S-NS)

---

**Input:** $\mathbf{x}$, **ncopies**, $n$
**Output:** $\mathbf{x}_{new}$, **ncopies**$_{new}$, **zeros**

1: $l \leftarrow 0, r \leftarrow n-1$
2: **for** $j \leftarrow 0; j < n; j \leftarrow j+1$ **do**
3:      **if ncopies**$^j > 0$ **then**
4:          **ncopies**$_{new}^l \leftarrow$ **ncopies**$^j$, $\mathbf{x}_{new}^l \leftarrow \mathbf{x}^j$, $l \leftarrow l+1$
5:      **else**
6:          **ncopies**$_{new}^r \leftarrow$ **ncopies**$^j$, $\mathbf{x}_{new}^r \leftarrow \mathbf{x}^j$, $r \leftarrow r-1$
7:      **end if**
8: **end for**
9: **zeros** $\leftarrow n-l$

---

**Algorithm 4** Rotational Split

---

**Input:** $\mathbf{x}$, **ncopies**, $N$, $P$, $n = \frac{N}{P}$, $p$
**Output:** $\mathbf{x}$, **ncopies**

1: $\mathbf{csum} \leftarrow \mathtt{Cumulative\_Sum}(N, P, \mathbf{ncopies})$
2: $\mathbf{min\_shifts}^j \leftarrow \mathbf{csum}^j - \mathbf{ncopies}^j - j - np$, $\forall j < n$ if $\mathbf{ncopies}^j > 0$; $-np$ makes $-j$ global
3: $\mathbf{max\_shifts}^j \leftarrow \mathbf{csum}^j - j - 1 - np$, $\forall j < n$ if $\mathbf{ncopies}^j > 0$
4: **for** $k \leftarrow 1; k \le \log_2 P; k \leftarrow k + 1$ **do** binary tree
5:　　$partner \leftarrow \left(p + \frac{P}{2^k}\right) \& (P - 1)$
6:　　**for** $j \leftarrow 0; j < n; j \leftarrow j + 1$ **do**
7:　　　　**if** $\mathbf{max\_shifts}^j \,\&\, N2^{-k} > 0$ **then**
8:　　　　　　**if** $\mathbf{min\_shifts}^j \,\&\, N2^{-k} > 0$ **then**
9:　　　　　　　　$\mathbf{copies\_to\_send}^j \leftarrow \mathbf{ncopies}^j$, $\mathbf{ncopies}^j \leftarrow 0$
10:　　　　　　**else**
11:　　　　　　　　$\mathbf{copies\_to\_send}^j \leftarrow (\mathbf{csum}^j - j - N2^{-k} - np)$
12:　　　　　　　　$\mathbf{ncopies}^j \leftarrow \mathbf{ncopies}^j - \mathbf{copies\_to\_send}^j$
13:　　　　　　**end if**
14:　　　　　　$starter \leftarrow \mathbf{csum}^j - \mathbf{copies\_to\_send}^j$, if $\mathbf{x}^j$ is the first particle to send
15:　　　　　　Send $\mathbf{x}^j$, $\mathbf{copies\_to\_send}^j$ to $partner$ and send $starter$ too if $\mathbf{x}^j$ is the first particle to send
16:　　　　**else**
17:　　　　　　Send 0s to $partner$ (Message to reject)
18:　　　　**end if**
19:　　**end for**
20:　　Accept or reject the received particles and $starter$, reset $starter$ to 0 if all particles are sent and none is accepted
21:　　$\mathbf{csum}^0 \leftarrow starter + \mathbf{ncopies}^0$, $\mathbf{csum}^j \leftarrow \mathbf{csum}^{j-1} + \mathbf{ncopies}^j$ $\forall j = 1, 2, ..., n - 1$
22:　　Update $\mathbf{min\_shifts}$ and $\mathbf{max\_shifts}$ as in steps 2 and 3
23: **end for**
24: **if** $P < N$ **then** perform leaf stage of the binary tree
25:　　**for** $j \leftarrow n - 1; j \ge 0; j \leftarrow j - 1$ **do**
26:　　　　**if** $\mathbf{csum}^j > (p + 1)n$ **then**
27:　　　　　　$\mathbf{copies\_to\_send}^j \leftarrow \min(\mathbf{csum}^j - (p + 1)n, \mathbf{ncopies}^j)$
28:　　　　　　$\mathbf{ncopies}^j \leftarrow \mathbf{ncopies}^j - \mathbf{copies\_to\_send}^j$
29:　　　　　　Send $\mathbf{x}^j$, $\mathbf{copies\_to\_send}^i$ to $partner$
30:　　　　**else**
31:　　　　　　Send 0s to $partner$ (Message to reject)
32:　　　　**end if**
33:　　　　**if** $\mathbf{min\_shifts}^j > 0$ **then**
34:　　　　　　Shift particle to the right by $\mathbf{min\_shifts}^j$
35:　　　　**end if**
36:　　**end for**
37:　　Accept or reject the received particles
38: **end if**

---

**Algorithm 5** Rotational Nearly Sort and Split (RoSS) Redistribution

---

**Input:** $\mathbf{x}$, **ncopies**, $N$, $P$, $n = \frac{N}{P}$, $p$
**Output:** $\mathbf{x}$

1: **if** $P > 1$ **then**
2:　　$\mathbf{x}$, **ncopies** $\leftarrow \mathtt{Rotational\_Nearly\_Sort}(\mathbf{x}, \mathbf{ncopies}, N, P, n, p)$, (9) now holds
3:　　$\mathbf{x}$, **ncopies** $\leftarrow \mathtt{Rotational\_Split}(\mathbf{x}, \mathbf{ncopies}, N, P, n, p)$, (11) now holds
4: **end if**
5: $\mathbf{x} \leftarrow \mathtt{S\text{-}R}(\mathbf{x}, \mathbf{ncopies}, n)$

### 4.1. General Overview

The algorithm consists of two phases. In the first phase, we want the elements in **ncopies** to be *nearly sorted*, a property which is defined as follows.

**Definition 2.** *A sequence of N non-negative integers,* **ncopies***, is nearly sorted in descending order when it has the following shape:*

$$\mathbf{ncopies} = \left[\lambda^0, \lambda^1, ..., \lambda^{m-1}, 0, ..., 0\right] \tag{9}$$

*where* $\mathbf{ncopies}^i > 0$ *(marked with $\lambda$s in (9)) $\forall i = 0, 1, ..., m-1$ and $0 \leq m \leq N$. On the other hand,* **ncopies** *is an ascending nearly sorted sequence if the last m elements are positive and the first are* 0*.*

In this paper, **ncopies** is nearly sorted in descending order. We also note that, as the elements in **ncopies** are progressively shifted to achieve (9), the related particles in **x** are consequently also shifted. The main purpose of this phase is to separate all particles that must be duplicated (i.e., those for which $\mathbf{ncopies}^i > 0$) from those that must be deleted. Here, we prove that (9) can be achieved by using Rotational Nearly Sort, an $O(\log_2 N)$ alternative to the $O((\log_2 N)^2)$ Nearly Sort step in [7] (as also described in Appendix A). We denote that (9) can also be achieved with sort, as carried out in [25] by using Bitonic Sort. In theory, one could also employ $O(\log_2 N)$ sorting networks [32,33]. However, the constant time of these networks is notoriously prohibitive in practice, even with some of the most recent advances [34]. Our constant time is significantly smaller; because of that, we do not consider AKS-like sorting networks as a practical approach for (9).

In the second phase, we want to achieve two goals: the first is to make room on the right of each particle that has to be copied; the second is for the $P$ cores to have the same number of particles in their private memory. The first goal easily translates to shifting the particles to the right until **ncopies** has the following new shape:

$$\mathbf{ncopies} = [\lambda^0, 0, ..., 0, \lambda^1, 0, ..., 0, \lambda^{m-1}, 0, ..., 0] \tag{10}$$

where, for each $\mathbf{ncopies}^i > 0$ (again marked with $\lambda$s in (10)), $\mathbf{ncopies}^i - 1$ zeros follow. The second can be expressed as follows:

$$\sum_{i=p\frac{N}{P}}^{(p+1)\frac{N}{P}-1} \mathbf{ncopies}^i = \frac{N}{P} \quad \forall p = 0, ..., P-1 \tag{11}$$

which is essentially (5) applied locally. In the next section, we prove it is possible to achieve both (10) and (11) in $O(\log_2 N)$ parallel time by using a single algorithm, which we refer to as Rotational Split. After that, the cores are completed by using S-R independently to redistribute the particles within their private memory.

### 4.2. Algorithmic Details and Theorems

In this section, we give details for a novel implementation of parallel redistribution and prove that it scales as $O(\log_2 N)$. The reader is referred to Figure 3, which illustrates an example for $N = 8$ and $P = 4$.

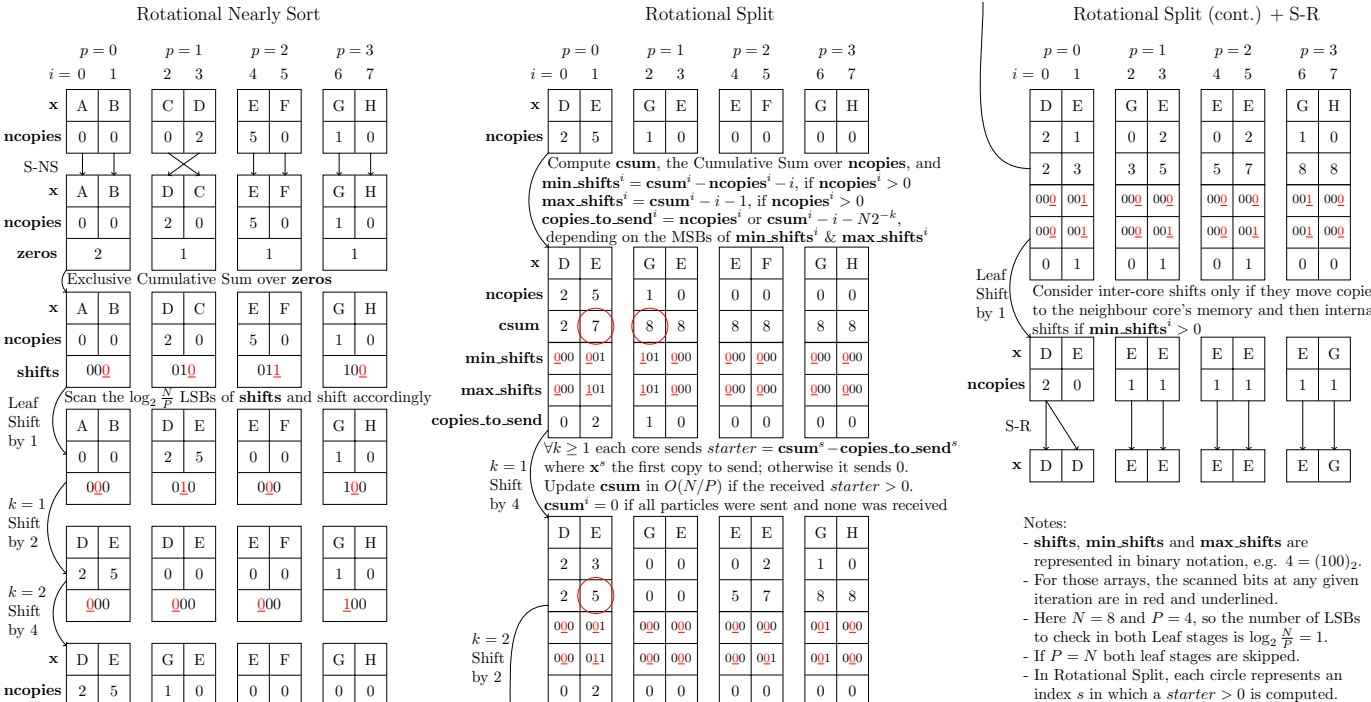

**Figure 3.** RoSS redistribution—example for $N = 8$ and $P = 4$. Each $\mathbf{x}^i$ is actually a real vector, but marked with a letter for brevity.

#### 4.2.1. Rotational Nearly Sort

**Theorem 1.** *Given an array of N particles, $\mathbf{x}$, and their copies to be created, **ncopies**, whose elements are evenly distributed across the P cores of a DM, Algorithm 2 (performed by each core p, $\forall p = 0, 1, ..., P - 1$) describes the steps to safely shift the elements in $\mathbf{x}$ and **ncopies** to achieve property (9), and performs that in $O(\log_2 N)$ parallel time for $P = N$.*

**Proof of Theorem 1.** The first process to carry out is to nearly sort the particles locally by calling Sequential Nearly Sort (S-NS) (see Algorithm 3), which iteratively moves the $i$-th particle to the left/right side of the output array if **ncopies**$^i$ is positive/zero. We note that this routine only takes $O(\frac{N}{P})$ iterations, since every DM core owns $\frac{N}{P}$ particles. This will start moving the particles locally to the left before doing so globally.

The particles within the core $p$ must now shift to the left by as many positions as the number of zero elements in **ncopies** owned by the cores with a lower rank. Let **zeros** $\in \mathbb{Z}^P$ be the array that counts the number of **ncopies**$^i = 0$ within each core; each element of **shifts** $\in \mathbb{Z}^P$ (the array to keep track of the remaining shifts) can be initialized as follows:

$$\mathbf{shifts}^p = \sum_{\hat{p}=0}^{p-1} \mathbf{zeros}^{\hat{p}} \tag{12}$$

Equation (12) can be parallelized by using a parallel exclusive cumulative sum (after each core $p$ has initialized **zeros**$^p$ to the sum of zeros within its memory, at the end of S-NS). It is well covered in the literature that the cumulative sum (also known as prefix sum or scan) takes $O(\frac{N}{P} + \log_2 P)$ computations [35,36].

We now want to express **shifts**$^p$ in binary notation and shift the particles by increasing power-of-two numbers of positions, depending on the bits of **shifts**$^p$, from the Least Significant Bit (LSB) to the Most Significant Bit (MSB). This translates to using a bottom-up binary tree structure that will complete the task in $O(\log_2 N)$ time.

If $P < N$, then we first need to perform an extra leaf stage, in which, the cores send all of the particles to their neighbor if the bitwise AND of **shifts**$^p$ and $\frac{N}{P} - 1$ is positive. In simple terms, the leaf stage masks the $\log_2 \frac{N}{P}$ LSBs of **shifts**$^p$ and performs the rotations referring to those bits in one operation per particle. Hence, during this step, each core sends and receives up to $\frac{N}{P}$ particles, meaning that the leaf stage takes $O(\frac{N}{P})$.

After the leaf stage, the actual tree-structure routine can start. At every $k$-th stage of the tree (for $k = 1, 2, ..., \log_2 P$), any core $p$ will send to its partner $p - 2^{k-1}$ all its particles (i.e., $\mathbf{x}$ and $\mathbf{ncopies}$) and $\mathbf{shifts}^p - \frac{N}{P}2^{k-1}$ (i.e., the number of remaining shifts after the current rotation) if the bitwise AND of $\mathbf{shifts}^p$ and $\frac{N}{P}2^{k-1}$ is positive; this corresponds to checking a new bit of $\mathbf{shifts}^p$, precisely the one which is significant at this stage. We note that, in order to balance the communication at all stages, the cores are set to send particles to reject (i.e., having $\mathbf{ncopies}^i = 0$) if the LSBs are 0. Hence, this phase takes $O(\frac{N}{P}\log_2 P)$ because each core sends and receives $\frac{N}{P}$ particles $\log_2 P$ times.

Before the leaf stage, the particles in the memory of core $p$ must shift at most from one end to the other end, i.e., by $\mathbf{shifts}^p \leq N - 1$ positions. Any non-negative integer $\mathbf{shifts}^p \leq N - 1$ can always be represented in base-2 by $\log_2 N$ bits. Therefore, each particle will be shifted $O(\log_2 N)$ times, each time by an increasing power-of-two number of positions, and, since $\mathbf{shifts}$ gets updated in $O(1)$, Algorithm 2 achieves an $O(\log_2 N)$ time complexity. Furthermore, as long as no particle to be copied collides with or gets past another one, Algorithm 2 always achieves (9), since the shifts to perform will progressively decrease, whereas (12) will always hold. Lemma 1 and Corollary 1 prove that collisions and overtaking (defined in the following definitions) can never occur. □

**Definition 3** (Collisions). *Let $\mathbf{x}^i$ be a particle having $\mathbf{ncopies}^i \geq 1$, and $\mathbf{x}^j$ be a particle having $\mathbf{ncopies}^j \geq 1$, with $j = i + dist$, where $0 < dist < N - 1$. A collision would occur if dist is a power-of-two number, $\mathbf{x}^j$ is rotating to the left by dist and $\mathbf{x}^j$ stays where it is. More formally, a collision occurs if the total number of rotations that $\mathbf{x}^j$ must perform has significant bit (i.e., the bit corresponding to dist rotations) equal to 1, while the same bit of the number of rotations that $\mathbf{x}^i$ must perform is 0. The same definition can be applied to collisions when rotations are performed to the right if $\mathbf{x}^i$ is rotating to the right (and, hence, have significant bit equal to 1) and $\mathbf{x}^j$ is not rotating (and, hence, have significant bit equal to 0).*

**Definition 4** (Overtaking). *Let $\mathbf{x}^i$ again be a particle having $\mathbf{ncopies}^i \geq 1$, and $\mathbf{x}^j$ again be a particle having $\mathbf{ncopies}^j \geq 1$, with $j = i + dist$, where $0 < dist < N - 1$. The particle $\mathbf{x}^j$ can overtake $\mathbf{x}^i$ if it is rotating to the left by a power-of-two number greater than dist while $\mathbf{x}^i$ stays where it is. The same problem occurs when rotations are performed to the right, if $\mathbf{x}^i$ is rotating to the right by a power-of-two number greater than dist while $\mathbf{x}^j$ stays where it is.*

**Lemma 1.** *During the $k$-th iteration of Rotational Nearly Sort, $\forall k = 1, 2, ..., \log_2 P$, a particle $\mathbf{x}^j$, having $\mathbf{ncopies}^j \geq 1$ and rotating to the left by $\frac{N}{P}2^{k-1}$ positions, can never collide with a particle $\mathbf{x}^i$, having $\mathbf{ncopies}^i \geq 1$ and $j = i + \frac{N}{P}2^{k-1}$.*

**Proof of Lemma 1.** At the $k$-th iteration, particle $\mathbf{x}^i$ has $\mathbf{shifts}^i$ remaining rotations to the left, whereas $\mathbf{x}^j$ has $\mathbf{shifts}^j$. Therefore, the necessary and sufficient condition for collisions, defined in Definition 3, can be restated for Rotational Nearly Sort by checking whether the following logical condition

$$\left( \left( \mathbf{shifts}^i \& \frac{N}{P}2^{k-1} \right) = 0 \right) \wedge \left( \left( \mathbf{shifts}^j \& \frac{N}{P}2^{k-1} \right) > 0 \right) \tag{13}$$

is true, which corresponds to checking whether the significant bit of $\mathbf{shifts}^i$ is 0 and the significant bit of $\mathbf{shifts}^j$ is 1. This condition can also be rearranged as follows:

$$\left( \left( \mathbf{shifts}^i \& \frac{N}{P}2^{k-1} \right) = 0 \right) \wedge \left( \left( (\mathbf{shifts}^i + \mathbf{zeros}^{i+1:j-1}) \& \frac{N}{P}2^{k-1} \right) > 0 \right) \tag{14}$$

where $\mathbf{zeros}^{i+1:j-1}$ is the number of 0s in $\mathbf{ncopies}$ between positions $i$ and $j$ excluded.

Since Rotational Nearly Sort performs rotations using an LSB-to-MSB strategy, it is easy to infer that the bits to the right of the significant one at this iteration (i.e., bit $k - 1$) are all 0. This means that, if the significant bit of $\mathbf{shifts}^i$ is 0, the only condition that would

make (14) true would be $\mathbf{zeros}^{i+1:j-1} = \frac{N}{P}2^{k-1}$. That is, however, impossible, because, in this case, there are only $i + \frac{N}{P}2^{k-1} - 1$ memory slots between $i$ and $j$, which means that:

$$\mathbf{zeros}^{i+1:j-1} \leq j - i - 1 = \frac{N}{P}2^{k-1} - 1 < \frac{N}{P}2^{k-1} \tag{15}$$

□

**Corollary 1.** *During the k-th iteration of Rotational Nearly Sort, $\forall k = 1, 2, ..., \log_2 P$, a particle $\mathbf{x}^j$, having $\mathbf{ncopies}^j \geq 1$ and rotating to the left by $\frac{N}{P}2^{k-1}$ positions, can never overtake a particle $\mathbf{x}^i$, having $\mathbf{ncopies}^i \geq 1$ and $i < j < i + \frac{N}{P}2^{k-1}$.*

**Proof of Corollary 1.** Lemma 1 can automatically prove Corollary 1 because, once again, (14) is true only if $\mathbf{zeros}^{i+1:j-1} = \frac{N}{P}2^{k-1}$. However, in this case, $j < i + \frac{N}{P}2^{k-1}$, and hence:

$$\mathbf{zeros}^{i+1:j-1} \leq j - i - 1 < j - i < \frac{N}{P}2^{k-1} \tag{16}$$

□

### 4.2.2. Rotational Split

**Theorem 2.** *Given an array of N particles, $\mathbf{x}$, and their copies to be created, $\mathbf{ncopies}$, whose elements are nearly sorted according to (9), and evenly distributed across the P cores of a DM, Algorithm 4 (performed by each core p, $\forall p = 0, 1, ..., P - 1$) describes the steps to shift and/or safely split the elements in $\mathbf{x}$ and $\mathbf{ncopies}$ to achieve properties (10) and (11), and performs that in $O(\log_2 N)$ parallel time for $P = N$.*

**Proof of Theorem 2.** Let $\mathbf{csum} \in \mathbb{Z}^N$ be the inclusive cumulative sum of $\mathbf{ncopies}$. As mentioned in the previous section, the cumulative sum achieves an $O(\frac{N}{P} + \log_2 P)$ time complexity for any $P \leq N$.

To achieve (10), we want to move the particles to the right to have $\mathbf{ncopies}^i - 1$ gaps after each index $i$, such that $\mathbf{ncopies}^i > 0$. It can be inferred that the minimum required number of shifts to the right that the $i$-th particle must perform is:

$$\mathbf{min\_shifts}^i = \sum_{z=0}^{i-1}(\mathbf{ncopies}^z - 1) = \mathbf{csum}^i - \mathbf{ncopies}^i - i \tag{17}$$

However, (10) alone does not guarantee (11). This is because, for each particle $\mathbf{x}^i$ that must be copied more than once, we are rotating all of its copies by the same minimum number of positions, and we are not considering that some of these copies could be split and rotated further to also fill in the gaps that a strategy for (10) alone would create. Therefore, we also consider the maximum number of rotations that any copy of $\mathbf{x}^i$ has to perform, without causing collisions. Since (10) aims to creating $\mathbf{ncopies}^i - 1$ gaps for each $i$, such that $\mathbf{ncopies}^i > 0$, that number is:

$$\mathbf{max\_shifts}^i = \mathbf{min\_shifts}^i + \mathbf{ncopies}^i - 1 = \mathbf{csum}^i - i - 1 \tag{18}$$

Therefore, the key idea is to consider $\mathbf{min\_shifts}^i$ and $\mathbf{max\_shifts}^i$ in binary notation and use their bits to rotate the particles accordingly, from the MSB to the LSB. This corresponds to using a top-down binary tree structure of $\log_2 P$ stages. At each stage $k$ of the tree (for $k = 1, 2, ..., \log_2 P$), any core with rank $p$ sends particles $N2^{-k}$ positions ahead to its partner with rank $p + \frac{P}{2^k}$. At each $k$-th stage, we check the MSB of both $\mathbf{min\_shifts}^i$ and $\mathbf{max\_shifts}^i$ to infer $\mathbf{copies\_to\_send}^i$, the number of copies of $\mathbf{x}^i$ that must rotate by $N2^{-k}$ positions. For each $\mathbf{x}^i$, three possible scenarios may occur:

- None of its copies must move;
- All of them must rotate;
- Some must split and shift, and the others must not move.

Trivially, the copies will not move if the MSB of $\mathbf{max\_shifts}^i$ is 0, which also implies that the MSB of $\mathbf{min\_shifts}^i$ is 0, since $\mathbf{min\_shifts}^i \leq \mathbf{max\_shifts}^i$ at all stages. If both MSBs are 1, we send $\mathbf{copies\_to\_send}^i = \mathbf{ncopies}^i$ copies of $\mathbf{x}^i$. However, if only the MSB of $\mathbf{max\_shifts}^i$ is equal to 1, $\mathbf{copies\_to\_send}^i < \mathbf{ncopies}^i$. The number of copies to split is equal to how many of them must be placed from position $i + N2^{-k}$ onwards to achieve a perfect workload balance. This is equivalent to computing how many copies make

$$\mathbf{csum}^i > i + N2^{-k}$$

Therefore, if only the MSB of $\mathbf{max\_shifts}^i$ is equal to 1, we send

$$\mathbf{copies\_to\_send}^i = \mathbf{csum}^i - i - N2^{-k} \tag{19}$$

copies of $\mathbf{x}^i$ and we keep the remaining ones where they were. Here, as in Section 4.2.1, particles to be rejected are sent if the MSB of $\mathbf{max\_shifts}^i$ is 0 in order to keep all cores equally busy. This phase takes an $O(\frac{N}{P} \log_2 P)$ time complexity, because each core sends or receives $\frac{N}{P}$ particles at every stage. However, to ensure a logarithmic time complexity, one needs to update $\mathbf{csum}$, $\mathbf{min\_shifts}$ and $\mathbf{max\_shifts}$ in $O(\frac{N}{P})$. This can be achieved if the cores send

$$starter = \mathbf{csum}^s - \mathbf{copies\_to\_send}^s \tag{20}$$

where $s$ is the index of the first particle to send, having $\mathbf{ncopies}^s > 0$. As long as no particle overwrites or moves past another one (which is proven to be impossible by Lemma 2), each core $p$ can safely see the received $starter$ as

$$starter = \sum_{i=0}^{p\frac{N}{P}-1} \mathbf{ncopies}^i = \mathbf{csum}^{p\frac{N}{P}-1}$$

and use it to re-initialize $\mathbf{csum}$ and update it in $O(\frac{N}{P})$ as $\mathbf{csum}^i = \mathbf{csum}^{i-1} + \mathbf{ncopies}^i$ $\forall i = p\frac{N}{P} + 1, \ldots, (p+1)\frac{N}{P} - 1$. This strategy guarantees $\mathbf{csum}$ is always correct for at least any index $i$, such that $\mathbf{ncopies}^i > 0$, i.e., those indexes we require to be correct. Once $\mathbf{csum}$ is updated, Equations (17) and (18) are embarrassingly parallel.

If $P < N$, after $\log_2 P$ stages, the cores perform a leaf stage. Here, for each particle, we perform inter-core shifts or splits and shifts only if there is any copy to be sent to the neighbor; this is equivalent to checking if

$$\mathbf{csum}^i > (p+1)\frac{N}{P} \tag{21}$$

where $(p+1)\frac{N}{P}$ is the first global index in the neighbor's memory. Arrays of 0s are again sent when no inter-core shifts are needed. We also consider internal shifts only if $\mathbf{min\_shifts}^i > 0$, in order to both make room to receive particles from the neighbor, $p - 1$, and guarantee (10). This leaf stage takes $O(\frac{N}{P})$ because $\frac{N}{P}$ particles are sent or received.

It is easy to infer that $\mathbf{min\_shifts}^i < N - 1$ and $\mathbf{max\_shifts}^i \leq N - 1$, as a particle copy could at most be shifted, or split and shifted from the first to the last position. Both $\mathbf{min\_shifts}^i$ and $\mathbf{max\_shifts}^i$ can be represented in base-2 by $\log_2 N$ bits. Therefore, as also anticipated above, only up to $\log_2 N$ messages are required, which means that the achieved time complexity of Algorithm 4 is $O(\log_2 N)$. Furthermore, the particles are progressively split according to the MSBs of $\mathbf{min\_shifts}^i$ and $\mathbf{max\_shifts}^i$, until they are split according to (21). Hence, after the final leaf stage, the first and last element of $\mathbf{csum}$ in the memory of each core will necessarily meet the following two requirements:

$$\left( \mathbf{csum}^{p\frac{N}{P}} - \mathbf{ncopies}^{p\frac{N}{P}} = p\frac{N}{P} \right) \wedge \left( \mathbf{csum}^{(p+1)\frac{N}{P}-1} = (p+1)\frac{N}{P} \right)$$

which, if subtracted, automatically proves (11). □

**Lemma 2.** *Given a nearly sorted input* **ncopies**, *at the k-th iteration of Rotational Split*, $\forall k = 1, 2, \ldots, \log_2 P$, *a particle* $\mathbf{x}^i$, *having* $\mathbf{ncopies}^i \geq 1$ *and rotating to the right by* $N2^{-k}$ *positions, can never collide with or get past a particle* $\mathbf{x}^j$, *having* $\mathbf{ncopies}^j \geq 1$ *and* $j = i + dist$ *with* $1 \leq dist \leq N2^{-k}$.

**Proof of Lemma 2.** This lemma can be proved in two possible complementary cases:

1. There are one or more zeros between *i* and *j*;
2. There are no zeros between *i* and *j*.

*Case 1.* Since the particles are initially nearly sorted, at the beginning, there are no zeros in between any pair of particles in position *i* and *j*. At the *k*-th iteration, if one or more zeros are found between $\mathbf{x}^i$ and $\mathbf{x}^j$, it necessarily means that $dist > \mathbf{max\_shifts}^i \geq N2^{-k}$. This is because of two reasons. First, zeros between two particles $\mathbf{x}^i$ and $\mathbf{x}^j$ can only be created if the MSB of $\mathbf{min\_shifts}^i$ is 0 and the MSB of $\mathbf{max\_shifts}^j$ is 1. Second, for any binary number, its MSB, if equal to 1, is a greater number than the one represented by any disposition of all remaining LSBs (e.g., $(1000)_2 = 8 > (0111)_2 = 7$). Hence, if there are any zeros between $\mathbf{x}^i$ and $\mathbf{x}^j$, it is because, during at least one of the previous stages, $\mathbf{x}^j$ rotated by an MSB and $\mathbf{x}^i$ did not, such that $\mathbf{x}^j$ is now beyond reach of possible collisions with $\mathbf{x}^i$.

*Case 2.* In this case, all particles between *i* and *j* are still nearly sorted. Therefore, $\mathbf{x}^i$ would collide with (when $dist = N2^{-k}$) or get past $\mathbf{x}^j$ (when $dist < N2^{-k}$) if

$$\left( \left( \mathbf{max\_shifts}^i \& N2^{-k} \right) > 0 \right) \wedge \left( \left( \mathbf{min\_shifts}^j \& N2^{-k} \right) = 0 \right) \tag{22}$$

is true, which corresponds to checking whether the MSB of $\mathbf{max\_shifts}^i$ is 1 (which also includes those cases where the MSB of $\mathbf{min\_shifts}^i$ is 1) and the MSB of $\mathbf{min\_shifts}^j$ is 0. In other words, (22) can be simplified to checking if $\mathbf{max\_shifts}^i > \mathbf{min\_shifts}^j$. However, for a pair of particles $\mathbf{x}^i$ and $\mathbf{x}^j$ within a nearly sorted group of particles, this is impossible, because:

$$\begin{aligned}
\mathbf{max\_shifts}^i &= \mathbf{csum}^i - i - 1 = \mathbf{csum}^j - \sum_{z=i+1}^{j} \mathbf{ncopies}^z - j + dist - 1 \\
&= \mathbf{csum}^j - \mathbf{ncopies}^j - j - \left( \sum_{z=i+1}^{j-1} \mathbf{ncopies}^z - dist + 1 \right) \\
&= \mathbf{min\_shifts}^j - \left( \sum_{z=i+1}^{j-1} \mathbf{ncopies}^z - dist + 1 \right) \leq \mathbf{min\_shifts}^j
\end{aligned}$$

since $\mathbf{csum}^j = \sum_{z=0}^{j} \mathbf{ncopies}^z$, $dist = j - i$ and (in this case) $\sum_{z=i+1}^{j-1} \mathbf{ncopies}^z \geq j - i - 1$. $\square$

### 4.2.3. Rotational Nearly Sort and Split Redistribution

**Theorem 3.** *Given an array of N particles*, $\mathbf{x}$, *and their copies to be created*, **ncopies**, *whose elements are evenly distributed across the P cores of a DM, Algorithm 5 (performed by each core p,* $\forall p = 0, 1, \ldots, P - 1$) *redistributes all particles in* $\mathbf{x}$ *for which* $\mathbf{ncopies}^i > 0$, *and performs that in* $O(\log_2 N)$ *parallel time for* $P = N$.

**Proof of Theorem 3.** After the cores perform Rotational Nearly Sort first, and then Rotational Split, Equation (11) holds, as proven by Theorem 2. Therefore, the cores can independently use S-R to redistribute their $\frac{N}{P}$ particle copies in their private memory in $O(\frac{N}{P})$ iterations. As proven by Theorems 1 and 2, Algorithms 2 and 4 complete their task in $O(\log_2 N)$ parallel time, which means that the achieved time complexity by Algorithm 5 is then $O(N)$ for $P = 1$, $O(\log_2 N)$ for $P = N$ cores and, for any $1 \leq P \leq N$, is:

$$O\left( \frac{N}{P} + \frac{N}{P} \log_2 P \right) \tag{23}$$

the first term in (23) represents S-R, which is always performed, and all of the steps that are only ever called once for any $P > 1$ (e.g., S-NS). The second term in (23) describes the

$\log_2 P$ stages of Algorithms 2 and 4, during which, we update, send and receive up to $\frac{N}{P}$ particles. □

With the results of Theorem 3 in view, we conclude that the PF tasks now take either $O(\frac{N}{P})$, $O(\frac{N}{P} + \log_2 P)$ or $O(\frac{N}{P} + \frac{N}{P} \log_2 P)$ time. This is because (3), (4) and (8) require reduction (which notoriously scales as $O(\frac{N}{P} + \log_2 P)$ [36]), the IS step is embarrassingly parallel and (in the resampling algorithm) Equation (6) requires an exclusive cumulative sum; in addition, Equation (7) and resetting the weights to $1/N$ are also embarrassingly parallel tasks. Table 1 summarizes this conclusion. This means that, even if we had an embarrassingly parallel fully-balanced redistribution, the time complexity of PFs will still be no less than $O(\frac{N}{P} + \log_2 P)$, since reduction and the cumulative sum are required elsewhere.

**Table 1.** Time complexity of each task of PFs on DM.

| Task Name (Parallelization Strategy) | Details | Sequential Time Complexity | Parallel Time Complexity |
|---|---|---|---|
| IS (embarrassingly parallel) | Equations (1) and (2) | $O(N)$ | $O(\frac{N}{P})$ |
| Normalize (reduction) | Equation (3) | $O(N)$ | $O(\frac{N}{P} + \log_2 P)$ |
| ESS (reduction) | Equation (4) | $O(N)$ | $O(\frac{N}{P} + \log_2 P)$ |
| MVR (cumulative sum) | Equations (6) and (7) | $O(N)$ | $O(\frac{N}{P} + \log_2 P)$ |
| Redistribution (RoSS) | Algorithm 5 | $O(N)$ | $O(\frac{N}{P} + \frac{N}{P} \log_2 P)$ |
| Reset (embarrassingly parallel) | $\mathbf{w}_t^i = 1/N \; \forall i$ | $O(N)$ | $O(\frac{N}{P})$ |
| Estimate (reduction) | Equation (8) | $O(N)$ | $O(\frac{N}{P} + \log_2 P)$ |

*4.3. Implementation on MPI*

In this section, we give brief information concering which MPI routines are needed to implement RoSS and the rest of SIR on MPI.

The exclusive cumulative sum that is required before the leaf stage in Algorithm 2 is parallelized on MPI by calling `MPI_Exscan` [37]. On the other hand, `MPI_Scan` is used to parallelize the inclusive cumulative sum of **ncopies** at the start of Algorithm 4.

During the binary tree and the leaf stages in Algorithms 2 and 4, the cores send and receive $\frac{N}{P}$ particles each time. On MPI, `MPI_Sendrecv` is ideal for these messages, since it requires the ranks to send to and receive from. Temporary arrays should be used on both communication ends to ensure data coherency before accepting or rejecting the received content.

All of the other operations or algorithms within RoSS, such as (17) or S-NS, are performed locally by each core and, therefore, do not need to call MPI routines.

For completeness, we point out that, in the other PF tasks, the reduction operation in (3), (4) and (8) is parallelized on MPI by calling `MPI_Allreduce`, whereas (6) needs `MPI_Exscan`.

## 5. Experimental Results

In this section, we show the numerical results of redistribution first and then show results for a PF example. In the experiment for redistribution, we compare RoSS, the novel fully-balanced algorithm presented in this paper, with N-R and B-R, two fully-balanced redistributions that take $O((\log_2 N)^2)$ steps (see Appendix A). These algorithms are compared by passing in input arrays with the same values to all algorithms: **ncopies** and **x** with $M = 1$. To guarantee (5), **ncopies** is generated randomly by using MVR (see Equation (7)) with a log-normally distributed input $\tilde{\mathbf{w}}$. For the PF experiment, we consider three versions of a SIR PF that only differ in terms of the constituent redistribution used. Each PF test is run for $T_{PF} = 100$ iterations. Resampling is computed every iteration so that we ensure that the frequency of redistribution is the same. The model we consider is the stochastic volatility example described in Appendix B.

All experiments are conducted for $N \in \{2^{16}, 2^{20}, 2^{24}\}$ particles and for up to $P = 256$ cores. Each reported run-time is the median of 20 runs collected for each $N$, $P$ pair.

All algorithms in this paper are coded in C++ with OpenMPI-1.10.7 and compiled with -O3 optimization flag; `double` and `unsigned int` data types are, respectively, used for real and integer numbers. Combinations of `MPI_Barrier` and `MPI_Wtime` are used to collect the run-times for Figures 4 and 5, while TAU Performance System 2.7 is used for the profiling in Figure 6. The cluster for the experiments consists of eight machines, interconnected with InfiniBand 100 Gbps and each mounting a 2 Xeon Gold 6138 CPU that provides 40 cores running at 2 GHz. We note that the results for up to $P = 32$ DM cores have been collected by requesting (at scheduling time) cores from the same node. For $P \geq 64$, we have requested 32 DM cores from each node. While we cannot identify any resulting artifacts in the results, we acknowledge that this feature of the hardware does potentially influence our results when comparing run-times related to $P \leq 32$ with those for $P \geq 64$. We emphasise that, to ensure the experimental results allow for a meaningful comparison between algorithms, all algorithms were assessed on the same hardware for each $P$. Future work would sensibly investigate these performance differences as part of a broader study on how to maximise performance as a function of hardware configuration.

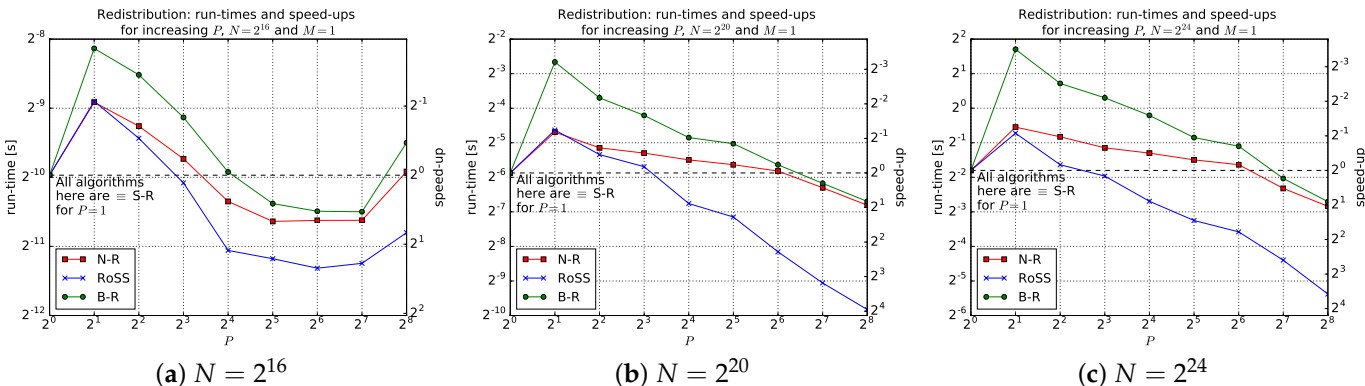

**Figure 4.** Redistribution—results for increasing $N$ and $P$.

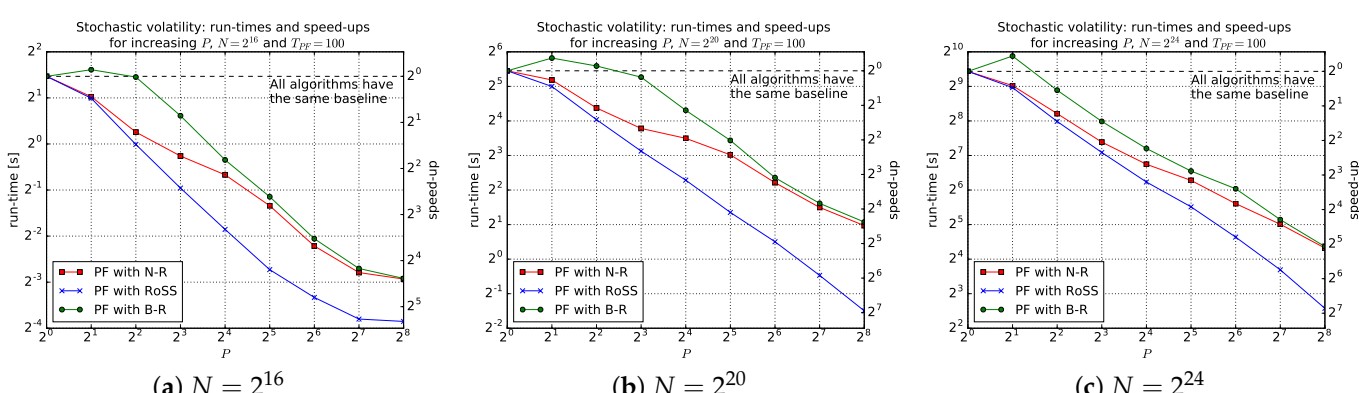

**Figure 5.** Stochastic volatility—results for increasing $N$ and $P$.

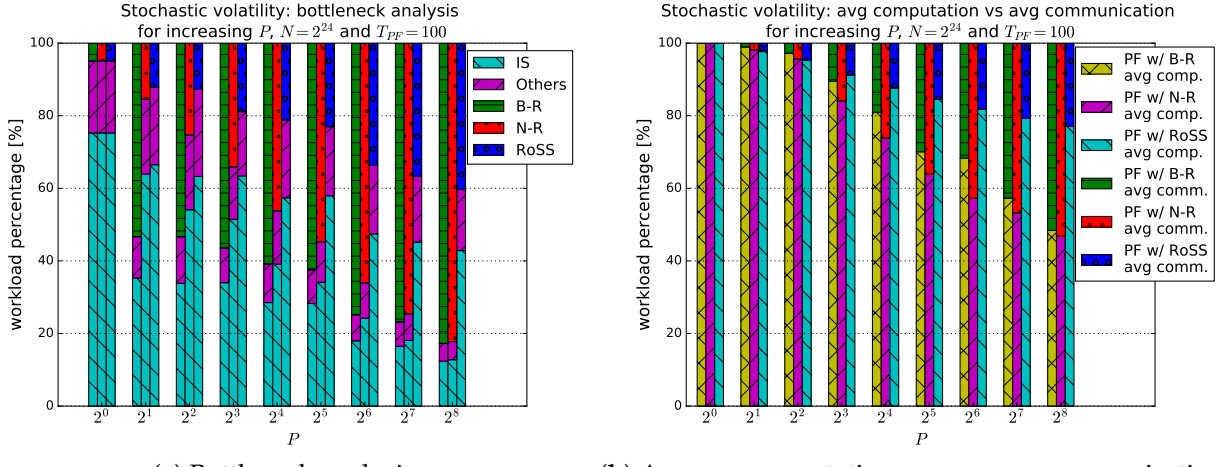

(**a**) Bottleneck analysis.  (**b**) Average computation vs. average communication.

**Figure 6.** Stochastic volatility—profiling information for $N = 2^{24}$ and increasing $P$.

### 5.1. RoSS vs. B-R and N-R

In Figure 4, we can see that the gap between the proposed approach and the other algorithms increases with $P$, particularly with large $N$. For $P \le 4$ and all three of the values for $N$ that are considered, RoSS is comparable with N-R and roughly four times as fast as B-R. However, for $P = 256$, RoSS is up to eight times faster than N-R (which is slightly faster than B-R, as shown in [7]) and B-R. We also note that these fully-balanced approaches may also stop scaling when the computation shrinks and the communication becomes too intensive: e.g., here, RoSS stops scaling at $N = 2^{16}$ and $P = 64$.

### 5.2. Stochastic Volatility

Figure 5 shows that the PF using RoSS is four to six times faster than a PF using N-R/B-R. The maximum speed-up for a PF with RoSS is approximately 125 for $P = 256$ cores. Furthermore, the gap between a PF with RoSS and a PF with any other considered redistribution also increases with $P$, in line with the results in Section 5.1. Figure 6 describes profiling information for the three PFs with $N = 2^{24}$. In Figure 6a, it is interesting to see that RoSS is the only redistribution variant that runs faster than IS for any $P \le 256$, while B-R and N-R emerge as the bottleneck for a relatively low $P$, i.e., for $2 \le P \le 16$. This is mostly due to the larger volume of communication in B-R/N-R, as shown in Figure 6b. In Figure 6b, we have considered the average time spent on all MPI calls per core as the communication time. Since $N \gg P$ in this experiment, the time spent on the `MPI_Sendrecv` calls in the redistribution step is dominant over the time spent on all the other MPI routines, such as `MPI_Allreduce` or `MPI_Exscan`, which only consists of $\log_2 P$ single-scalar messages and up to $2 \log_2 P$ arithmetical operations (in this case sum). Therefore, we can state that the reported average communication is equivalent (with a good degree of confidence) to the percentage of time that the cores are not used for any computation.

### 6. Conclusions

In this paper, we present RoSS, a novel fully-balanced redistribution for global resampling in SIR PFs on distributed memory environments. The algorithm has been implemented on MPI. The baselines for comparison are B-R and N-R, two similar state-of-the-art fully-balanced redistribution algorithms that both achieve an $O((\log_2 N)^2)$ time complexity and whose implementation is already available on MPI. We prove in this paper that RoSS redistribution achieves an $O(\log_2 N)$ time complexity.

The empirical results show that RoSS redistribution is almost an order of magnitude faster than B-R and N-R for up to $P = 256$ cores. Similar results are observed in the context of an exemplar PF. For the same level of parallelism, a PF using RoSS is up to six times faster than a PF using B-R/N-R and provides a maximum speed-up of 125. We also denote

that, under the same testing conditions, RoSS is the only option for redistribution that can still be faster than IS for such large $P$.

Future work should focus on reducing the number of messages between the cores. One way to achieve this is to combine SM with DM: we note that an $O(\log_2 N)$ redistribution for SM already exists [19], and that mixing OpenMP, one of the most common programming models for SM architectures, with MPI is a routine practice in the HPC domain [38]. A second avenue for potential improvement consists of using recent versions of OpenMP that support GPU offload or using Cuda (in place of OpenMP) in order to take advantage of the extra speed-up that GPU cards can offer relative to a CPU.

**Author Contributions:** Conceptualization, A.V. and S.M.; formal analysis, A.V., P.G.S. and S.M.; software, investigation, validation, visualization and writing original draft, A.V.; review and editing, A.V., P.G.S. and S.M.; supervision, P.G.S. and S.M.; project administration and funding acquisition, S.M. All authors have read and agreed to the published version of the manuscript.

**Funding:** This work was supported by a UK EPSRC Doctoral Training Award (1818639), Schlumberger, and the UK EPSRC "Big Hypotheses" (EP/R018537/1) and "Algorithmic Aspects of Temporal Graphs" (EP/P02002X/1) grants.

**Institutional Review Board Statement:** Not application.

**Informed Consent Statement:** Not application.

**Data Availability Statement:** Not application.

**Conflicts of Interest:** The authors declare no conflict of interest.

## Appendix A. $O((\log_2 N)^2)$ Fully-Balanced Redistribution

This appendix describes the fully-balanced redistribution algorithm described in [30]. This routine redistributes the particles in $O((\log_2 N)^2)$ parallel time by performing the following steps.

First, the particles **x** are sorted by the values in **ncopies**. The chosen sorting algorithm is Bitonic Sort, a comparison-based parallel sorting algorithm that has been implemented on a cluster of graphics cards relatively recently [28]. For any $P \leq N$, Bitonic Sort performs

$$O\left( \frac{N}{P} \left( \log_2 \left( \frac{N}{P} \right) \right)^2 + \frac{N}{P} (\log_2 P)^2 \right) \tag{A1}$$

comparisons, where the first term describes the number of steps to perform Bitonic Sort locally, and the second term represents the data movement to merge the keys between the cores. Here, as in [25], Bitonic Sort is performed locally, but it is also possible to replace it with an $O(N \log_2 N)$ single-core sort, such as Mergesort. Quicksort is not recommended in this case, because **ncopies** contains many zeros, which often results in Quicksort's worst-case ($O(N^2)$) run-time. An example of a sorting network for Bitonic Sort is illustrated in Figure A1. The particles to be duplicated are then separated from those to be deleted.

After this single sort, the algorithm moves on to another top-down routine. Starting from the root node, at each stage of the binary tree, three parallel operations are performed in sequence. First, the cores compute the inclusive cumulative sum over **ncopies**. Then, they search for a *pivot* that perfectly divides the node into two balanced leaves; in other words, *pivot* is the first index where the cumulative sum is equal to or greater than $\frac{N}{2}$. To find and broadcast *pivot* to all cores of the node, the cores use linear search locally, followed by a sum, which can be performed in parallel by using reduction. In the end, the $\frac{N}{2}$ particles on the right side of the *pivot* are shifted to the right side of the node. This is achieved by expressing $r$ (the positions to shift by) in base-2 and rotating in $O(\log_2 N)$ steps according to the bits of $r$. In this way, the root node gets split into two balanced leaves. This top-down routine is recursively performed $\log_2 P$ times until the workload is equally distributed across the cores; then, S-R is called. In this paper, as in [7], we refer to this algorithm as Bitonic-Sort-Based Redistribution (B-R). Since Bitonic Sort is used once,

and cumulative sum, reduction and rotational shifts are performed $\log_2 P$ times each, B-R achieves an $O((\log_2 N)^2)$ time complexity for $P = N$, or equal to (A1) for any $P \leq N$. As mentioned in Section 4.1, it would be theoretically possible to replace Bitonic Sort with AKS sort in order to make the asymptotic time complexity of the sorting phase equal to $O(\log_2 N)$, albeit with a large constant. However, since B-R needs a cumulative sum, a sum and rotational shifts up to $\log_2 N$ times, the overall computational complexity that would result from such an approach would still be $O((\log_2 N)^2)$. In [30], B-R has been implemented on MapReduce. In [25], B-R has been ported to MPI.

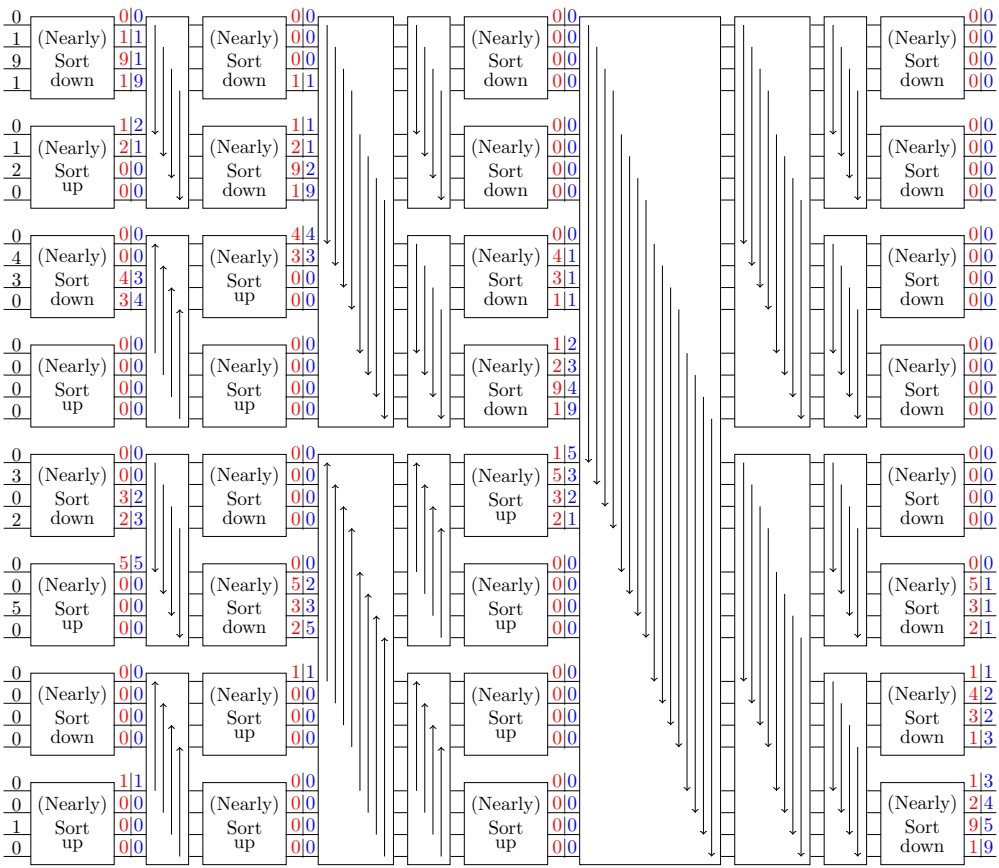

**Figure A1.** Bitonic/Nearly Sort for $N = 32$ and $P = 8$. The arrows represent inter-core messages (e.g., `MPI_Sendrecv`) and swap, which, for Nearly Sort, only applies to a zero-positive pair, since pairs of positive keys or zeros are nearly sorted. After each stage, results are in blue/red for Bitonic/Nearly Sort.

A small change to B-R can result in a further 25% improvement, as results in [7] and Section 5.1 show. This is achieved by substituting Bitonic Sort with an alternative algorithm, Nearly Sort. One does not actually need to perfectly sort the particles to divide the workload afterwards, but only needs to guarantee **ncopies** has shape (9). To achieve this property, the particles are first nearly sorted locally by calling Algorithm 3. We emphasize that doing so is faster than sorting the particles according to **ncopies**. Then, the particles are recursively merged, as in Bitonic Sort, by using the same sorting network illustrated in Figure A1. Since S-NS takes $O(\frac{N}{P})$, and the number of sent/received messages per core equals $(\log_2 P)^2$, we can infer that Nearly Sort costs

$$O\left(\frac{N}{P} + \frac{N}{P}(\log_2 P)^2\right) \qquad (A2)$$

iterations. Here, as in [7], we refer to a B-R parallelization that uses Nearly Sort instead of Bitonic Sort as Nearly-Sort-Based Redistribution (N-R). Algorithm A1 summarizes both

N-R and B-R, which achieve an $O((\log_2 N)^2)$ time complexity. Figure A2 shows an example of N-R for $N = 16$ and $P = 8$; a figure for B-R is omitted due to its similarities with N-R.

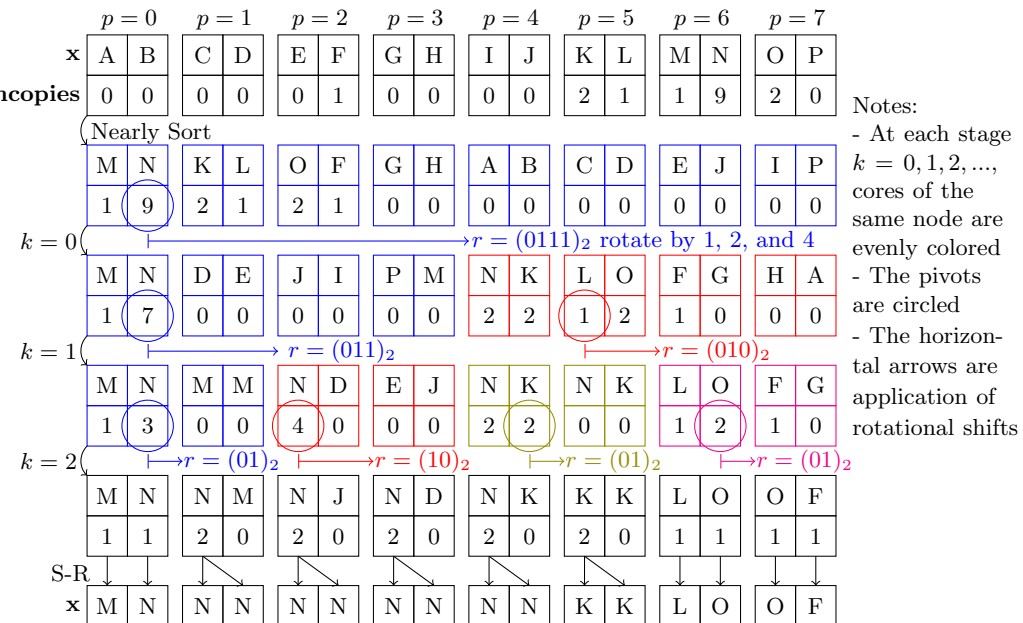

**Figure A2.** N-R - example for $N = 16$ and $P = 8$. Each $\mathbf{x}^i$ is actually a real vector, but is marked with a capital letter for brevity.

---

**Algorithm A1** Bitonic/Nearly-Sort-Based Redistribution (B-R/N-R)

---

**Input:** $\mathbf{x}$, **ncopies**, $N$, $P$, $n = \frac{N}{P}$, $p$
**Output:** $\mathbf{x}$

1: **if** $P > 1$ **then** Bitonic/Nearly Sort the particles
2:      `Bitonic/Nearly_Sort`(**ncopies**, $\mathbf{x}$, $N$, $P$)
3: **end if**, **ncopies** has now shape (9)
4: **for** $k \leftarrow 0; k < \log_2 P; k \leftarrow k + 1$ **do** binary tree
5:      $\mathbf{csum} \leftarrow$ `Cumulative_Sum`$\left(\frac{N}{2^k}, \frac{P}{2^k}, \mathbf{ncopies}\right)$, the $\frac{P}{2^k}$ cores in each node perform
         cumulative sum over **ncopies**
6:      $pivot \leftarrow$ `Linear_Search`$(\mathbf{ncopies}, \mathbf{csum}, n)$, search for first $pivot$ s.t. $\mathbf{csum}^{pivot} \geq \frac{N}{2^{k+1}}$;
         if not found $pivot \leftarrow 0$
7:      $pivot \leftarrow$ `Sum`$(pivot, \frac{P}{2^k}, p)$, the $\frac{P}{2^k}$ cores in each node broadcast $pivot$ to each other
8:      $r \leftarrow \frac{N}{2^{k+1}} - pivot$, rotations to perform within the node
9:      $\mathbf{ncopies}, \mathbf{x} \leftarrow$ `Rot_Shifts`$\left(\mathbf{ncopies}, \mathbf{x}, r, pivot, \frac{P}{2^k}, p\right)$, the $\frac{P}{2^k}$ cores in each node rotate
         the $\frac{N}{2^{k+1}}$ particles on the right of $pivot$ by $r$ positions according to the bits of $r$
10: **end for**, **ncopies** has now shape (11)
11: $\mathbf{x} \leftarrow$ `S-R`$(\mathbf{x}, \mathbf{ncopies}, n)$

---

## Appendix B. Stochastic Volatility Model

The PF example considered in Section 5.2 is a stochastic volatility model that has previously appeared in the literature [15,19] and estimates the GBP-to-USD exchange rate from 1 October 1981 to 28 June 1985. The model is as follows:

$$\mathbf{X}_t = \phi\mathbf{X}_{t-1} + \sigma\mathbf{V}_t \tag{A3a}$$

$$\mathbf{Y}_t = \beta \exp(0.5\mathbf{X}_t)\mathbf{W}_t \tag{A3b}$$

where $\phi = 0.9731$, $\sigma = 0.1726$, $\beta = 0.6338$ (as selected in [19]). $\mathbf{V}_t \sim \mathcal{N}(0, 1)$ and $\mathbf{W}_t \sim \mathcal{N}(0, 1)$, which means that $p(\mathbf{x}_t^i|\mathbf{x}_{t-1}^i)$ and $p(\mathbf{Y}_t|\mathbf{x}_t^i)$ in (2) are also Gaussian. The initial

state is sampled as $\mathbf{X}_0 \sim \mathcal{N}(0, \frac{\sigma^2}{1-\phi^2})$. The particles are initially drawn from $p_0(\mathbf{X}_0)$ and then from the dynamic model. Hence, (2) simplifies to $\mathbf{w}_t^i = \mathbf{w}_{t-1}^i p\left(\mathbf{Y}_t | \mathbf{x}_t^i\right)$.

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
