# Peer review of "An O(log2N) Fully-Balanced Resampling Algorithm for Particle Filters on Distributed Memory Architectures"

_algorithms, doi:10.3390/a14120342_

Round 1
Reviewer 1 Report
This submission presents a novel algorithm for a particular kind of data redistribution that is periodically required in a distributed memory algorithm for particle filters; this algorithm has lower complexity than previous algorithms for this purpose. The paper is very well written with a clear description of the background and mathematical basis of the work, of the algorithm itself with a precise analysis of its correctness and complexity, and an experimental evaluation of an MPI implementation of the algorithm that demonstrates its practical superiority over previous approaches.
Thus I essentially recommend the acceptance of the paper in its present form; however (as a minor recommendation), Figure 3 should be shifted to the beginning of Section 4.2 where it is first referenced.
Reviewer 2 Report
The authors present a parallel redidistribution implementation (RoSS) for distributed memory architectures. Their proposal is evaluated using set of parameters (i.e. problem size, number of cores) and they also compare his proposal with fully balanced resdistrubutions such as N-R and B-R. They also present an interesting performance analysis but it would be interesting to see the results of tests for speedup and utilization of cores.
I recommend the authors to add a link to a public repository with the code of the proposed parallel implementations.
After revising the manuscript, I am recommending the acceptance subject to minor changes.
Reviewer 3 Report
In this paper, starting from the title, the authors propose a low time-complexity algorithm for Fully Balanced Resampling on distributed memory machines in the context of Particle Filters. The motivation resides in the execution time of this component in real-time applications, even if the distributed memory architecture is unusual for real-time context. The proposed algorithm is interesting, renews concepts from bitonic sort, and introduces a mechanism to reduce message passing among processing units.
The paper focuses on the parallel time complexity, which is an insidious topic as it strongly depends on the specific topology of the considered machine. The authors present their work sometimes overlooking the number of processing units, here improperly called cores, as the total number P rarely appears in the presented complexities. A reader can suppose they are implicitly assuming an N-PRAM machine, but at that point, the context of a distributed machine appears unclear.
Moreover, communication overhead should receive particular attention when talking about parallel algorithms on distributed machines, measuring its impact on the execution time. The authors barely mention this topic within the text. In general, if the contribution of this paper is on the parallel performance of the algorithm on a distributed memory architecture, some innovative algebraic approaches to the performance analysis are available in the literature.
Other improvable points per section are:
- In section 2, eq.(6), cdf is defined in terms of ncopies, and in eq.(7), ncopies is defined using cdf, while (7) seems to follow (6) in the sequential algorithm. Eq.(6) deserves a better contextualization. In referred literature, cdf appears defined in terms of w.
- In section 3, the last sentence seems out of context, as P=N would be unrealistic in a Distributed Memory architecture, creating overwhelming inefficiency due to communication overhead.
- In section 4:
- the authors present several algorithm segments using a singular illustrative figure positioned in section 5. A better organization would ease the reading on digital support;
- the unusual presentation of the algorithms using theorems to describe the algorithmic steps seems unmotivated. Moreover, supporting lemmas usually precede the theorem to facilitate the comprehension of the proof. The authors should ponder the overall readability of the section;
- table 1 presents a list of parallel time complexities for the required tasks in the case of P=N. This hypothesis is not stated in the description and seems unrealistic in the context of Distributed Memory architectures. The reader could consider the entire evaluation as useless for the actual case.
- In section 5, the distributed memory cluster used for the experiment actually has a hybrid shared/distributed memory architecture. The authors use the message passing paradigm also on a single node with 40 processing cores. Consequently, presented results reflect this setting, are less general than supposed, and do not consider the communication overhead properly. Finally, the authors do not present a speed-up analysis, typical in the performance analysis of a scalable parallel algorithm.
Round 2
Reviewer 3 Report
The authors significantly improved their paper.
The paragraph preceding table 1 is still presenting asymptotic complexities without explicit mention. The only hint is the final hypothesis of the previous theorem. The reader could get confused, considering the following summarizing table.